# Thermal Accelerometer Simulation by the R-Functions Method

**Mikhail Basarab** [1,*] , **Alain Giani** [2] **and Philippe Combette** [2]

[1] Computer Science and Control Systems Department, Bauman Moscow State Technical University, 105005 Moscow, Russia
[2] Institut d'Electronique et des Systèmes (IES), Université de Montpellier, 34095 Montpellier, CEDEX 05, France; alain.giani@ies.univmontp2.fr (A.G.); philippe.combette@ies.univ-montp2.fr (P.C.)
* Correspondence: bmic@mail.ru; Tel.: +7-903-760-5499

**Abstract:** As well as many modern devices, thermal accelerometers (TAs) need a sophisticated mathematical simulation to find the ways for their performance optimization. In the paper, a novel approach for solving computational fluid dynamics (CFD) problems in the TA's cavity is proposed (MQ-RFM), which is based on the combined use of Rvachev's R-functions method (RFM) and the Galerkin technique with multiquadric (MQ) radial basis functions (RBFs). The semi-analytical RFM takes an intermediate position between traditional analytical approaches and numerical methods, such as the finite-element method (FEM), belonging to the family of the so-called meshless techniques which became popular in the last decades in solving various CFD problems in complex-shaped cavities. Mathematical simulation of TA by using the MQ-RFM was carried out with the purpose to simulate the temperature response of the device and to study and improve its performance. The results of numerical experiments were compared with well-known analytical and numerical benchmark solutions for the circular annulus geometry and it demonstrated the effectiveness of the MQ-RFM for solving the convective heat-transfer problem in the TA's cavity. The use of solution structures allows one to take a relatively small number of expansion terms to achieve an appropriate accuracy of the approximate solution satisfying at the same time the given boundary conditions exactly. The application of the MQ-RFM gives the possibility to obtain semi-analytical solutions to the diffusion-convection problems and to identify the main thermal characteristics of the TA, that allows one to improve the device performance.

**Keywords:** natural convection; Galerkin method; R-functions method; multiquadric RBF; thermal accelerometer; cylindrical annulus

## 1. Introduction

Many modern devices, in particular, sensors based on microelectromechanical systems (MEMS), need a sophisticated mathematical simulation to find the ways of their optimization. Among them are thermal accelerometers (TAs) [1] which principle of operation is based on the effect of fluid or gas convection in closed cavities under the influence of external forces of inertia. It is very important to estimate the scale factor and bias stabilities of TAs under an external thermal slope, and to test different types of cavity geometry (cylindrical, rectangular, hexagonal, etc.) to achieve the best performance of the device.

To solve the aforementioned problems, it is necessary to simulate diffusive and convective heat and mass transfer in arbitrarily shaped enclosures, which in turn is a problem of great importance due to a large number of its practical applications. Since analytical solutions exist for a very narrow class of simple-shaped domains only, a majority of modern computational fluid dynamics (CFD) techniques

were developed which are based mainly on the use of the finite-difference and finite- element methods (FDM and FEM) [2–4], boundary-element methods [5], and spectral element methods [6]. Their main drawback is the cumbersome representation of grid solutions in complex- shaped domains, as well as the difficulties connected with the approximation of boundary conditions. For the function of pressure or for the vorticity function in dimensionless form, boundary conditions are not specified at all and require special approximations. In addition, sometimes it is difficult to interpret analytically the solutions obtained at a number of mesh knots and to evaluate errors.

Recently, a number of meshless (or mesh-free) techniques were proposed for solving convection-diffusion problems with arbitrarily shaped cavities [7–9]. They became popular due to their simplicity, flexibility, and independence from a complicated domain geometry. One of the most effective meshless techniques is the Rvachev's R-function method (RFM) [10]. It allows the given boundary conditions to be satisfied exactly at all boundary points by means of the appropriate transformation of the basic functions. With R-functions, it became possible to construct the functions with prescribed values and derivatives at specified locations. Furthermore, the constructed functions possess desired differential properties and may be assembled into a solution structure for the posed boundary value problems. The semi-analytical method of R-functions takes an intermediate position between traditional analytical approaches and numerical tools. Many practical problems in different areas of mathematical physics, in particular, heat conductivity [11], are being solved effectively by using RFM.

Application of the RFM in combination with the Galerkin technique for solving CFD problems in arbitrarily shaped domains was investigated earlier [12–14]. Different bases were applied in these works, both spectral (polynomial) and local (B-splines), and good results were achieved for fields evaluated in domains of simple geometry without localized inhomogeneities. In [15], an approach combining the RFM and the Petrov–Galerkin method with bases of algebraic polynomials was presented and applied to the simulation of thermal convection fields inside a closed rectangular cavity.

However, when applying the RFM, if a domain consists of two parts, for example, "inner" and "outer", which characteristic dimensions are not comparable, one needs to take either a large higher power of approximating polynomials (in spectral methods) that worsens computational stability, or a very small regular or irregular mesh width for bases of compactly supported functions. Therefore, in the latter case, the RFM loses its main benefits and can be practically considered as a conventional grid method such as FDM or FEM.

Here, a novel technique is proposed (MQ-RFM), based on the RFM with multiquadric (MQ) radial basis functions (RBFs) [16,17]. These functions are very popular as "building blocks" in different meshless methods intended for solving CFD problems [18,19]. Multiquadrics have a simple algebraic representation and, like algebraic or trigonometric polynomials, are spectral ones, i.e., not compactly supported. On the other hand, the basis of MQs can be simply formed on a usual regular mesh in the way similar to the bases of compactly supported bilinear or high-order splines. For the first time, the idea of using the RFM in combination with RBFs for solving linear boundary value problems for partial differential equations (PDEs) was proposed in [20], where irregular mesh was constructed on the base of the Voronoi diagrams.

In this work, the standard model of circular annulus geometry is studied which earlier was extensively investigated experimentally [21–23], analytically [24,25], and numerically [26–33]. This model is the basic one for studying the TA behavior and performance [34–36]. The distance between the heater and detectors was found providing an optimum sensitivity of the sensor.

The results of numerical experiments confirm the effectiveness of RFM applied for solving the convective heat-transfer problem in the TA's cavity and also demonstrate good accordance with the results obtained experimentally, analytically, and numerically with the help of FEM simulation [34,36]. The use of RFM solution structures and MQ-RBFs defined on a simple regular mesh allows one to take a relatively small number of expansion terms to achieve an appropriate accuracy of the approximate solution satisfying the given boundary conditions exactly. The application of the MQ-RFM gives a possibility to obtain the semi-analytical solutions to the diffusion-convection problems and identify

main thermal characteristics of the TA of more complicated geometry [37], that allows one to improve the device performance.

## 2. Thermal Accelerometer and Its Principle of Operation

The principle of operation of a TA is based on the effect of fluid or gas convection in a closed cavity under the influence of external forces of inertia. The device (Figure 1) includes a heating element H that creates around itself a symmetrical thermal field. The thermal sensors are located on opposite sides of this element.

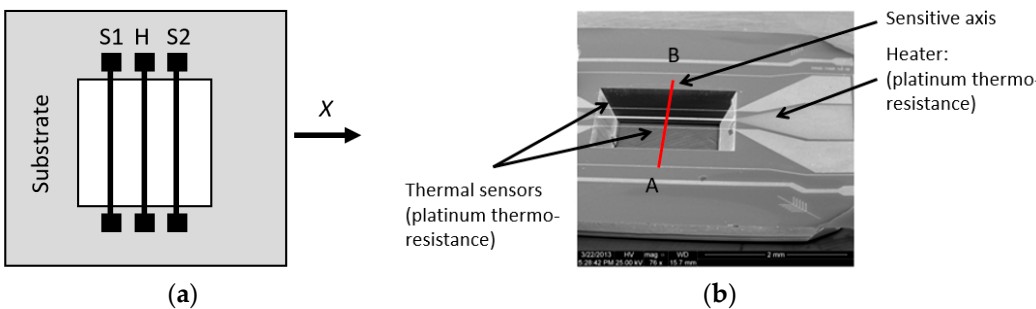

**Figure 1.** The general scheme (**a**) and microstructure design (**b**) of a thermal accelerometers (TA).

In the absence of an external acceleration along the sensitive axis $X$, the system is balanced and the heat detectors generate the same signal (Figure 2a). In presence of an external acceleration, the temperature profile changes, which results in the temperature difference $dT$ between the sensors S1 and S2, depending on the amplitude of the acceleration (Figure 2b). Nonzero temperature difference between the heat sensors converts the input impedance to the output electrical signal.

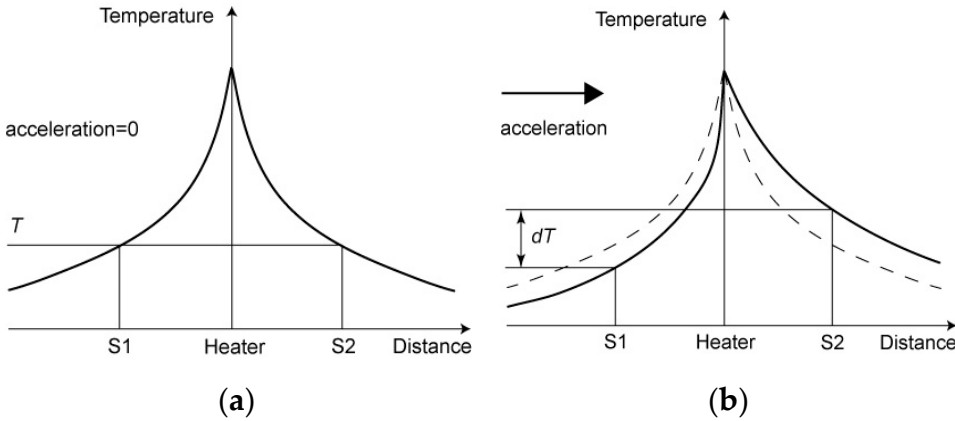

**Figure 2.** The principle of operation of a TA. (**a**) without acceleration, (**b**) with acceleration.

This type of accelerometer has several important advantages over conventional sensors for acceleration based on microelectromechanical systems (MEMS). In particular, due to the absence of moving mechanical parts, convective accelerometers have high reliability, low cost of production, as well as the ability to withstand and measure high loads caused, for example, by shock action.

To find the optimal parameters of the device: the type of gas and its pressure, the size and geometry of the cavity, the material and the size of the heating element and heat sensors, etc., it is necessary to solve the boundary value problem described by a system of the Navier–Stokes differential equations. This problem can be solved analytically only for a limited class of simple regions (cylinder, sphere), and in the general case, a numerical simulation is required.

## 3. Statement of the Problem and Governing Equations

Usually, the temperature distribution inside the 3D cavity of the device is almost uniform in the direction along the heater and sensors. Thus we need to solve the 2D heat-transfer problem in the middle cross-section of the cavity. One of the most important models of the TA is the circular annulus model (Figure 3a) [34,36] because for such a domain the analytical solution to the heat diffusion problem is known as well as the asymptotic solution of the diffusion-convection problem [24,25].

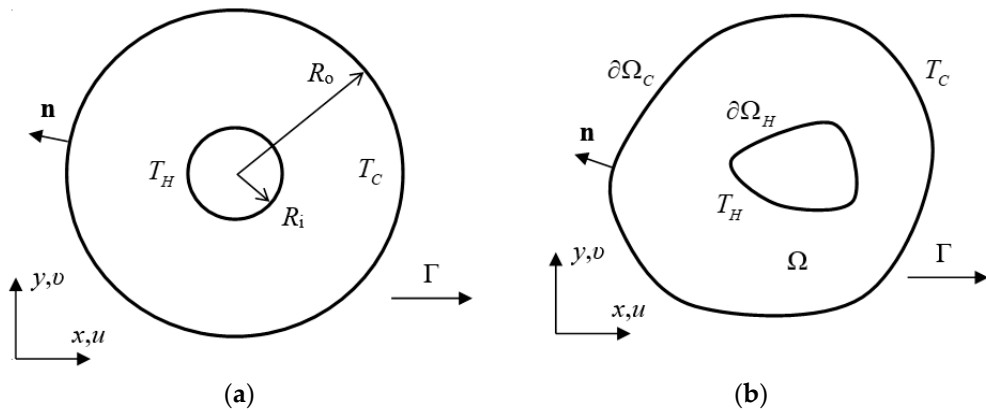

**Figure 3.** Convection-diffusion problem in a circular annulus (**a**) and geometry of a common boundary-value problem (**b**).

We consider the Boussinesq approximation with the assumption of the steady state incompressible flow inside an arbitrary 2D closed cavity $\Omega \subset \mathbf{R}^2$ (Figure 3b), when all constants, except density, do not depend on temperature. Without loss of generality, we take the domain with boundary composed of two parts: $\partial\Omega_C$, $\partial\Omega_H$, with given constant temperatures $T_C$ and $T_H$ on each of them respectively. Let external acceleration $\Gamma$ be applied towards the positive direction of sensitivity axis $OX$.

The dimensionless governing equations for the two-dimensional convective heat and mass transfer inside $\Omega$ have the following form:

$$
\begin{aligned}
&\tfrac{1}{\Pr}\nabla^2\theta - \left(U\tfrac{\partial\theta}{\partial X} + V\tfrac{\partial\theta}{\partial Y}\right) = 0; \\
&\nabla^2\zeta - \left(U\tfrac{\partial\zeta}{\partial X} + V\tfrac{\partial\zeta}{\partial Y}\right) = -\mathrm{Gr}\tfrac{\partial\theta}{\partial Y}; \\
&\nabla^2\psi = -\zeta;\ U = \tfrac{\partial\psi}{\partial Y},\ V = -\tfrac{\partial\psi}{\partial X},
\end{aligned}
\tag{1}
$$

where $\theta = (T - T_C)/(T_H - T_C)$ is the dimensionless temperature; $T$ is the temperature; $\psi$ is the stream function; $\zeta$ is the vorticity; $X = x/L$, $Y = y/L$ are dimensionless coordinates; $L$ is the specific dimension; $U = uL/\nu$; $V = vL/\nu$ are dimensionless components of velocity; u, v are horizontal and vertical components of velocity; $\nu$ is the kinematic viscosity; $\mathrm{Gr} = \Gamma\beta(T_H - T_C)L^3/\nu^2$ is the Grashof number; $\Gamma$ is the acceleration; $\beta$ is the temperature volume expansion coefficient; $\Pr = c_p\mu/\lambda$ is the Prandtl number; $\mu$ is the dynamic viscosity; $\lambda$ is the heat conductivity; $c_p$ is the specific heat at constant pressure.

Therefore, on $\partial\Omega$ the following dimensionless boundary conditions are given:

$$
\begin{aligned}
&\partial\Omega_C :\ \theta = 0; \\
&\partial\Omega_H :\ \theta = 1; \\
&\partial\Omega :\ \psi = \tfrac{\partial\psi}{\partial\mathbf{n}} = U = V = 0,
\end{aligned}
\tag{2}
$$

where $\mathbf{n}$ is the outward normal to boundary $\partial\Omega$.

Boundary conditions for the vorticity function $\zeta$ are not defined explicitly and are usually approximated by its Taylor series expansion in the neighborhood of the boundary.

## 4. Generalized Numerical Procedure

First of all, before applying common variational techniques considered in [38,39], we should pass from inhomogeneous boundary conditions (2) to homogeneous ones by introducing the new function:

$$\widetilde{\theta} = \theta - \Phi, \tag{3}$$

where $\Phi$ is a continuous function which is equal to unity on the part $\partial\Omega_H$ and vanishes on $\partial\Omega_C$. Function $\Phi$ can be constructed in the way similar to that for the traditional Lagrange interpolation polynomial but here instead of interpolation points we take corresponding segments of the boundary. This technique also is called the transfinite interpolation [40].

Hereinafter, to avoid cumbersome designations, we shall denote:

$$\theta \equiv \widetilde{\theta}.$$

Thus, we get:

$$\frac{1}{\Pr}\nabla^2\theta - \left(U\frac{\partial\theta}{\partial X} + V\frac{\partial\theta}{\partial Y}\right) = -\frac{1}{\Pr}\nabla^2\Phi + U\frac{\partial\Phi}{\partial X} + V\frac{\partial\Phi}{\partial Y};$$
$$\nabla^2\zeta - \left(U\frac{\partial\zeta}{\partial X} + V\frac{\partial\zeta}{\partial Y}\right) = -\mathrm{Gr}\cdot\left(\frac{\partial\theta}{\partial Y} + \frac{\partial\Phi}{\partial Y}\right); \tag{4}$$
$$\nabla^2\psi = -\zeta;\ U = \frac{\partial\psi}{\partial Y},\ V = -\frac{\partial\psi}{\partial X};$$

$$\partial\Omega:\ \theta = \psi = \frac{\partial\psi}{\partial\mathbf{n}} = U = V = 0. \tag{5}$$

Equations of System (4) can be solved iteratively:

$$\frac{1}{\Pr}\nabla^2\theta^{(k)} - \left(U^{(k-1)}\frac{\partial\theta^{(k)}}{\partial X} + V^{(k-1)}\frac{\partial\theta^{(k)}}{\partial Y}\right) = -\frac{1}{\Pr}\nabla^2\Phi + U^{(k-1)}\frac{\partial\Phi}{\partial X} + V^{(k-1)}\frac{\partial\Phi}{\partial Y};$$
$$\nabla^2\zeta^{(k)} - \left(U^{(k-1)}\frac{\partial\zeta^{(k)}}{\partial X} + V^{(k-1)}\frac{\partial\zeta^{(k)}}{\partial Y}\right) = -\mathrm{Gr}\left(\frac{\partial\theta^{(k)}}{\partial Y} + \frac{\partial\Phi}{\partial Y}\right); \tag{6}$$
$$\nabla^2\psi^{(k)} = -\zeta^{(k)};\ U^{(k)} = \frac{\partial\psi^{(k)}}{\partial Y},\ V^{(k)} = -\frac{\partial\psi^{(k)}}{\partial X};\ (k = 1, 2, \ldots).$$

Initial approximations $\zeta^{(0)}$, $\psi^{(0)}$, $U^{(0)}$, $V^{(0)}$ must be given. If $\zeta^{(0)} = \psi^{(0)} = U^{(0)} = V^{(0)} = 0$, then after the first step ($k = 1$) we obtain the solution to the stationary heat diffusion problem without convection.

The iterative process (6) stops when some convergence conditions are fulfilled, for example,

$$\frac{\left\|\theta^{(k+1)} - \theta^{(k)}\right\|_{\mathbf{L}_2(\Omega)}}{\left\|\theta^{(k+1)}\right\|_{\mathbf{L}_2(\Omega)}} \le \varepsilon \ll 1. \tag{7}$$

In a number of variational and projection techniques [6,38,39], approximations to $\theta^{(k)}$ at each iterative step are found in the form of truncated generalized Fourier series with respect to functions of some basis $\{f_n\}_{n=0}^{N}$:

$$\theta^{(k)} \approx \sum_{n=0}^{N} c_n^{(k)} f_n, \tag{8}$$

where $\mathbf{c}^{(k)} = (c_0^{(k)}, c_1^{(k)}, \ldots, c_N^{(k)})$ are undefined coefficients.

In addition to their differentiability, all functions $f_n$ must vanish on the boundary, i.e.,

$$\partial\Omega:\ f_n = 0,\ n = 0, \ldots, N. \tag{9}$$

From Equation (9) it follows that approximation (8) at any iterative step will also satisfy the homogeneous boundary condition (5) for $\theta$ exactly.

Leonid Kantorovich [38] proposed the following technique for constructing a functional basis satisfying Equation (9). First we should take an arbitrary basis system of functions $\{\chi_n\}_{n=0}^{N}$. There may be algebraic or trigonometric polynomials, splines, etc. Then a function $\omega(X, Y)$ is constructed such that $\omega > 0$ inside $\Omega$; $\omega < 0$ outside $\Omega \cup \partial\Omega$; $\omega = 0$ and $|\nabla\omega|_2 \neq 0$ on $\partial\Omega$. For simple domains, for example circular, these functions are trivial, but the common approach to obtaining analogous expressions for arbitrary domains was developed by V. Rvachev [10]. This approach (the R-functions method—RFM) will be considered in the next section.

Then we take the new system of functions

$$f_n \equiv \omega\chi_n, \ n = 0, \ldots, N, \tag{10}$$

which, due to the properties of the function $\omega$, satisfy the conditions (9).

Taking into account Equation (9), Equation (8) can be written as:

$$\theta^{(k)} \approx \sum_{n=0}^{N} c_n^{(k)} f_n = \omega \sum_{n=0}^{N} c_n^{(k)} \chi_n. \tag{11}$$

To strictly obey conditions (5) for the stream function, we must take another basis $\left\{\omega^2\chi_n\right\}_{n=0}^{N}$ with undefined coefficients $\mathbf{e}^{(k)} = (e_0^{(k)}, e_1^{(k)}, \ldots, e_N^{(k)})$:

$$\psi^{(k)} \approx \sum_{n=0}^{N} e_n^{(k)} \omega f_n = \omega^2 \sum_{n=0}^{N} e_n^{(k)} \chi_n. \tag{12}$$

Then the velocity components will be expressed as

$$U^{(k)} = \sum_{n=0}^{N} e_n^{(k)} \frac{\partial(\omega f_n)}{\partial Y}, \ V^{(k)} = -\sum_{n=0}^{N} e_n^{(k)} \frac{\partial(\omega f_n)}{\partial X}. \tag{13}$$

Substituting Equation (12) into the left-hand side of the third equation of system (4), we obtain a vorticity function expansion:

$$\zeta^{(k)} \approx \sum_{n=0}^{N} d_n^{(k)} \nabla^2(\omega f_n). \tag{14}$$

The latter expression may be considered as a representation of the vorticity function with respect to the set of basic functions $\nabla^2(\omega f_n)$ with a set of unknown coefficients $\mathbf{d}^{(k)} = (d_0^{(k)}, d_1^{(k)}, \ldots, d_N^{(k)})$:

$$d_n^{(k)} = -e_n^{(k)}, \ n = 0, \ldots, N. \tag{15}$$

Thus, it is necessary to solve recurrently only two equations of System (6) to find coefficients $c_n^{(k)}$ and $d_n^{(k)}$. It should also be noted here that Equation (14) provides us an expression for the vorticity functions without the necessity to approximate additionally its boundary values.

At each iteration, the realization of various numerical techniques requires different procedures of reducing an original problem to a corresponding system of linear algebraic equations (SLAE) with respect to unknown vectors $\mathbf{c}^{(k)}$ and $\mathbf{d}^{(k)}$. Here we propose to use an approach on the base of the Petrov–Galerkin procedure [6].

At the first step, we find the components $c_n^{(k)}$ by the Galerkin technique. Substituting Equation (11) into the first equation of (6), we obtain the residual:

$$\delta_\theta^{(k)} = \sum_{n=0}^{N} c_n^{(k)} \left( \frac{1}{\text{Pr}} \nabla^2 f_n - U^{(k-1)} \frac{\partial f_n}{\partial X} - V^{(k-1)} \frac{\partial f_n}{\partial Y} \right) - F_\theta^{(k-1)}. \tag{16}$$

Here the right-hand part is

$$F_\theta^{(k-1)} = -\left(\frac{1}{\text{Pr}}\nabla^2\Phi - U^{(k-1)}\frac{\partial\Phi}{\partial X} - V^{(k-1)}\frac{\partial\Phi}{\partial Y}\right).$$

One must choose the set of coefficients $c_n^{(k)}$ minimizing residual (16). In the Galerkin scheme, the orthogonality of $\delta_\theta$ to all functions of the system $\{f_n\}_{n=0}^N$ in space $\mathbf{L}_2(\Omega)$ is required, i.e.,

$$\int_\Omega \delta_\theta^{(k)} f_m \mathrm{d}\sigma = 0, \ m = 0,\ldots,N, \tag{17}$$

where $\mathrm{d}\sigma \equiv \mathrm{d}X\mathrm{d}Y$.

Substituting Equation (16) in Equation (17), we get the SLAE with respect to elements of the vector $\mathbf{c}^{(k)}$:

$$(\mathbf{A} + \overset{\smile}{\mathbf{A}}^{(k-1)})\mathbf{c}^{(k)} = \mathbf{b} + \overset{\smile}{\mathbf{b}}^{(k-1)}, \tag{18}$$

where components of matrices $\mathbf{A}, \overset{\smile}{\mathbf{A}}^{(k-1)}$ and vectors $\mathbf{b}, \overset{\smile}{\mathbf{b}}^{(k-1)}$ are determined respectively as:

$$\begin{aligned}
a_{m,n} &= \tfrac{1}{\text{Pr}}\int_\Omega f_m\nabla^2 f_n\mathrm{d}\sigma = -\tfrac{1}{\text{Pr}}\int_\Omega \nabla f_m\nabla f_n\mathrm{d}\sigma, \\
\overset{\smile}{a}_{m,n}^{(k-1)} &= -\int_\Omega f_m\left(U^{(k-1)}\tfrac{\partial f_n}{\partial X} + V^{(k-1)}\tfrac{\partial f_n}{\partial Y}\right)\mathrm{d}\sigma; \\
b_m &= -\tfrac{1}{\text{Pr}}\int_\Omega f_m\nabla^2\Phi\, dXdY = \tfrac{1}{\text{Pr}}\int_\Omega \nabla f_m\nabla\Phi\, \mathrm{d}\sigma, \\
\overset{\smile}{b}_m^{(k-1)} &= -\int_\Omega f_m\left(U^{(k-1)}\tfrac{\partial\Phi}{\partial X} + V^{(k-1)}\tfrac{\partial\Phi}{\partial Y}\right)\mathrm{d}\sigma; \\
& \qquad m,n = 0,\ldots,N.
\end{aligned} \tag{19}$$

Some optimization of Equation (19) can be done to decrease the time for computations.

Then we pass to finding coefficients of expansion (14). Analogously to the previous case, we write the corresponding residual for the second equation of the System (6) as:

$$\delta_\zeta^{(k)} = \sum_{n=1}^N d_n^{(k)}\left(\nabla^2\nabla^2(\omega f_n) - U^{(k-1)}\frac{\partial}{\partial X}\nabla^2(\omega f_n) - V^{(k-1)}\frac{\partial}{\partial Y}\nabla^2(\omega f_n)\right) - F_\zeta^{(k)}, \tag{20}$$

where

$$F_\zeta^{(k)} = -\text{Gr}\cdot\left(\frac{\partial\theta^{(k)}}{\partial Y} + \frac{\partial\Phi}{\partial Y}\right).$$

To find coefficients $d_n^{(k)}$, we make an orthogonal projection of the residual (20) to all functions of the basis $\{f_n\}_{n=0}^N$ that is the main idea of the Petrov–Galerkin method [6]:

$$\int_\Omega f_m\delta_\zeta^{(k)}\mathrm{d}\sigma = 0, \ m = 0,\ldots,N. \tag{21}$$

Finally we obtain the following SLAE for $\mathbf{d}^{(k)}$:

$$(\mathbf{W} + \overset{\smile}{\mathbf{W}}^{(k-1)})\mathbf{d}^{(k)} = \mathbf{z} + \overset{\smile}{\mathbf{z}}^{(k)}, \tag{22}$$

where

$$w_{m,n} = \int_\Omega f_m \nabla^2 \nabla^2 (\omega f_n)\, d\sigma;$$

$$\overset{\smile}{w}_{m,n}^{(k-1)} = -\int_\Omega f_m \Big( U^{(k-1)} \frac{\partial}{\partial X} \nabla^2 (\omega f_n) + V^{(k-1)} \frac{\partial}{\partial Y} \nabla^2 (\omega f_n) \Big) d\sigma;$$

$$z_m = -\mathrm{Gr} \int_\Omega f_m \frac{\partial \Phi}{\partial Y}\, d\sigma;$$

$$\overset{\smile}{z}_m^{(k)} = -\mathrm{Gr} \int_\Omega f_m \frac{\partial \theta^{(k)}}{\partial Y}\, d\sigma, \quad m, n = 1, \dots, N. \tag{23}$$

Integrals in Equations (19) and (23) can be evaluated numerically, and differential operators are approximated with their finite-difference analogs.

## 5. R-Functions Method and Transfinite Interpolation

The RFM [10] allows all prescribed boundary conditions to be satisfied exactly at all boundary points. The R-functions are real-valued functions that behave as continuous analogs of logical Boolean functions. With R-functions, it became possible to construct functions with prescribed values and derivatives at specified locations. Furthermore, the constructed functions possess desired differential properties and may be assembled into a solution structure that is guaranteed to contain solutions to the posed boundary value problems. For example, the homogeneous Dirichlet conditions may be satisfied exactly by representing the solution as the product of two functions: (i) a real-valued R-function that takes on zero values at the boundary points; and (ii) an unknown function that allows one to satisfy (exactly or approximately) the differential equation of the problem.

A function $f(x, y)$ is called an *R-function* if its sign is completely determined by the signs (but not magnitudes) of its arguments.

The most popular system of R-functions is the system with R-operations (R-conjunction, R-disjunction, and R-negation) defined in the following way:

$$\begin{aligned} x \wedge y &\equiv x + y - \sqrt{x^2 + y^2}, \\ x \vee y &\equiv x + y + \sqrt{x^2 + y^2}, \\ \overline{x} &\equiv -x. \end{aligned} \tag{24}$$

The above R-functions correspond to the Boolean logic functions $\wedge$, $\vee$, $\neg$ in a piecewise sense and allow constructing normalized implicit functions for complex-shaped geometric objects. Let the geometric domain $\Omega = B(\Omega_C, \Omega_H)$ be constructed as a Boolean (union and intersection) combination of primitive regions $\Omega_C, \Omega_H$, defined by real-valued functional inequalities $\omega_C(x, y) > 0$, $\omega_H(x, y) > 0$ respectively. If $f$ is an R-function corresponding to the Boolean function $B$, then the implicit function of the resulting geometric domain is immediately given by $\Omega = [f(\omega_C, \omega_H) > 0]$. The function $f(\omega_C, \omega_H)$ is negative outside of $\Omega$ and the equation $f(\omega_C, \omega_H) = 0$ defines the boundary $\partial\Omega$ of the domain $\Omega$.

It is known that the equation of the boundary $\partial\Omega$ ($f = 0$) is called normal if the value of $f(x, y)$ is equal to the Euclidean distance from the point $(x, y)$ to the boundary $\partial\Omega$. Similarly a function f that coincides with the normal function only on the boundary $\partial\Omega$ is called normalized and has a property that:

$$\left. \frac{\partial f}{\partial \mathbf{n}} \right|_{\partial\Omega} = -1. \tag{25}$$

If both implicit functions, $\omega_C, \omega_H$, are normalized on the boundaries $\Omega_C, \Omega_H$ respectively then all of the R-functions above preserve this property, and the function $f(\omega_C, \omega_H)$ is normalized on the whole boundary $\partial\Omega$.

With the help of R-functions, it is possible to make the transfinite interpolation, i.e., to construct a continuous expression satisfying different conditions on boundary parts $\partial\Omega_C$, $\partial\Omega_H$. Here, the "bonding" operation for boundary conditions must be used. For example, the function:

$$\Phi = \frac{\theta_H\omega_C + \theta_C\omega_H}{\omega_C + \omega_H} \tag{26}$$

satisfies the conditions

$$\Phi|_{\partial\Omega_C} = \theta_C, \ \Phi|_{\partial\Omega_H} = \theta_H. \tag{27}$$

The latter expression is a generalized analog of the Lagrange interpolation formula and allows us to pass from inhomogeneous boundary conditions to homogeneous ones.

The detailed justification of the RFM with Galerkin technique in connection with solving the natural convection problem in enclosure regions is presented in [12,41,42]. In particular, the natural convection in presence of local heat is investigated in [12]. This justification is based on variational principles [43] and is appropriate for the wide class of bases composed of both spectral and compactly supported functions.

It is convenient to take normalized functions due to the fact that with them it is easier to evaluate some important differential characteristics of the solution, in particular, the Nusselt number, Nu. The local Nusselt number at a point of boundary $\partial\Omega$ is expressed as:

$$\text{Nu}|_{\partial\Omega} = -\frac{\partial\theta}{\partial\mathbf{n}}\bigg|_{\partial\Omega}. \tag{28}$$

Taking into account Equations (3) and (11) and properties of the normalized function $\omega$, we can write that:

$$\text{Nu}|_{\partial\Omega} = -\frac{\partial}{\partial\mathbf{n}}\left(\omega\sum_{n=0}^{N}c_n^{(k)}\chi_n + \Phi\right)\bigg|_{\partial\Omega} = -\left(\frac{\partial\omega}{\partial\mathbf{n}}\sum_{n=0}^{N}c_n^{(k)}\chi_n + \omega\sum_{n=0}^{N}c_n^{(k)}\frac{\partial\chi_n}{\partial\mathbf{n}} + \frac{\partial\Phi}{\partial\mathbf{n}}\right)\bigg|_{\partial\Omega} = \left(\sum_{n=0}^{N}c_n^{(k)}\chi_n - \frac{\partial\Phi}{\partial\mathbf{n}}\right)\bigg|_{\partial\Omega}. \tag{29}$$

Substituting Equation (26) into Equation (29) and taking into account properties of the normalized functions $\omega_C, \omega_H$, we get:

$$\frac{\partial\Phi}{\partial\mathbf{n}}\bigg|_{\partial\Omega} = \frac{\partial}{\partial\mathbf{n}}\left(\frac{\theta_H\omega_C + \theta_C\omega_H}{\omega_C + \omega_H}\right)\bigg|_{\partial\Omega} = \frac{1}{\omega_C + \omega_H}\left[\left(\theta_H\frac{\partial\omega_C}{\partial\mathbf{n}} + \theta_C\frac{\partial\omega_H}{\partial\mathbf{n}}\right) - \Phi\left(\frac{\partial\omega_C}{\partial\mathbf{n}} + \frac{\partial\omega_H}{\partial\mathbf{n}}\right)\right]\bigg|_{\partial\Omega}.$$

Therefore on each part of the boundary $\partial\Omega$ we have the simple expressions:

$$\frac{\partial\Phi}{\partial\mathbf{n}}\bigg|_{\partial\Omega_C} = (\theta_C - \theta_H)\frac{1}{\omega_H}\bigg|_{\partial\Omega_C}, \ \frac{\partial\Phi}{\partial\mathbf{n}}\bigg|_{\partial\Omega_H} = (\theta_H - \theta_C)\frac{1}{\omega_C}\bigg|_{\partial\Omega_H}, \tag{30}$$

and

$$\text{Nu}|_{\partial\Omega_C} = \sum_{n=0}^{N}c_n^{(k)}\chi_n|_{\partial\Omega_C} + (\theta_H - \theta_C)\frac{1}{\omega_H}\bigg|_{\partial\Omega_C}, \ \text{Nu}|_{\partial\Omega_H} = \sum_{n=0}^{N}c_n^{(k)}\chi_n|_{\partial\Omega_C} + (\theta_C - \theta_H)\frac{1}{\omega_C}\bigg|_{\partial\Omega_H}. \tag{31}$$

Average Nusselt numbers on boundaries $\partial\Omega_C$, $\partial\Omega_H$ are found by integration of local Nusselt numbers along corresponding contours:

$$\overline{\text{Nu}}|_{\partial\Omega_C} = \frac{1}{|\partial\Omega_C|}\int_{\partial\Omega_C}\text{Nu}ds, \ \overline{\text{Nu}}|_{\partial\Omega_H} = \frac{1}{|\partial\Omega_H|}\int_{\partial\Omega_H}\text{Nu}ds. \tag{32}$$

## 6. Numerical Experiment

Consider the model boundary value problem in circular annulus (Figure 3a) with boundary conditions:

$$X^2 + Y^2 = R_i: \ \theta = 1, \ \psi = \frac{\partial \psi}{\partial \mathbf{n}} = U = V = 0;$$
$$X^2 + Y^2 = R_o: \ \theta = \psi = \frac{\partial \psi}{\partial \mathbf{n}} = U = V = 0. \tag{33}$$

The common way to obtain a function describing the boundary of the domain shown in Figure 3b is the use of R-conjunction operation:

$$\omega = \omega_C \wedge \omega_H = \omega_C + \omega_H - \sqrt{\omega_C^2 + \omega_H^2}. \tag{34}$$

For the circular annulus, expressions for "hot" and "cold" parts of the boundary can be written as follows:

$$\omega_H(X, Y) = -\frac{1}{2R_i}\left(R_i^2 - X^2 - Y^2\right), \ \omega_C(X, Y) = \frac{1}{2R_o}\left(R_o^2 - X^2 - Y^2\right). \tag{35}$$

Here, the normalizing terms $1/(2R_{i/o})$ are taken to provide the normalized functions $\omega_C, \omega_H$ whose values, as well as values of $\omega$, are close to the distances to corresponding boundaries at points located in their vicinity.

At first glance, for such a simple domain as shown in Figure 3a, by analogy with the approach proposed in [38,39], one would take instead of Equation (34) another expression:

$$\omega = \omega_C \omega_H = -\left(R_i^2 - X^2 - Y^2\right)\left(R_o^2 - X^2 - Y^2\right). \tag{36}$$

However, this way of constructing the function $\omega$ is restricted by the narrow class of domains (eccentric circular annuli, elliptic annuli, etc.) and it is inappropriate, for example, for annular domains with more complicated inner boundaries (triangular, rectangular, et al. [44,45]). Additionally, due to the fact that the function (36) is not normalized, namely,

$$\left.\frac{\partial \omega}{\partial \mathbf{n}}\right|_{\partial \Omega_C} = \omega_H|_{\partial \Omega_C}, \ \left.\frac{\partial \omega}{\partial \mathbf{n}}\right|_{\partial \Omega_H} = \omega_C|_{\partial \Omega_H},$$

we shall obtain more complicated formulae for the local Nusselt numbers than Equation (31) while the application of functions (35) yields the following expressions:

$$\text{Nu}|_{\partial \Omega_C} = \sum_{n=0}^{N} c_n^{(k)} \chi_n|_{\partial \Omega_C} + 2R_o \frac{\theta_H - \theta_C}{R_o^2 - R_i^2}, \ \text{Nu}|_{\partial \Omega_H} = \sum_{n=0}^{N} c_n^{(k)} \chi_n|_{\partial \Omega_H} + 2R_i \frac{\theta_C - \theta_H}{R_o^2 - R_i^2};$$
$$\overline{\text{Nu}}\Big|_{\partial \Omega_C} = \sum_{n=0}^{N} c_n^{(k)} \int_{\partial \Omega_C} \chi_n ds + 2R_o \frac{\theta_H - \theta_C}{R_o^2 - R_i^2}, \ \overline{\text{Nu}}\Big|_{\partial \Omega_H} = \sum_{n=0}^{N} c_n^{(k)} \int_{\partial \Omega_H} \chi_n ds + 2R_i \frac{\theta_C - \theta_H}{R_o^2 - R_i^2}.$$

With the help of the transfinite interpolation (26), we pass to the problem with homogeneous boundary conditions:

$$\Phi = \frac{\omega_C}{\omega_H + \omega_C}. \tag{37}$$

In this work, we take the basis of multiquadric RBF (MQ-RBF) which is often used in various meshless techniques [18,19]:

$$\chi_n(X, Y) = \sqrt{\left(X - X_{i(n)}\right)^2 + \left(Y - Y_{j(n)}\right)^2 + \alpha^2}, \tag{38}$$

where $\left(X_{i(n)} Y_{j(n)}\right)$ are the centers of functions $\chi_n$ and $\alpha$ is their shape parameter. For simplicity we take $\left(X_{i(n)} Y_{j(n)}\right)$ as knots of a regular two-dimensional square mesh with widths $h_X = h_Y = \alpha$.

Instead of the latter representation we may construct 2D MQ-RBF by means of the tensor product:

$$\chi_n(X, Y) = \sqrt{\left(X - X_{i(n)}\right)^2 + \alpha^2} \cdot \sqrt{\left(Y - Y_{j(n)}\right)^2 + \alpha^2}. \tag{39}$$

We used both formulae, (38) and (39), and numerical experiments demonstrated that from the point of view of computation costs and accuracy they are almost equivalent.

To define the appropriate parameters of our technique, we tested it on the problem of the stationary heat diffusion problem in the annulus cavity, whose stationary axisymmetric analytical solution is:

$$T(r) = T_C + (T_H - T_C)\left(1 - \frac{\ln\left(R_i^{-1}\right)}{\ln\left(R_o R_i^{-1}\right)}\right). \tag{40}$$

Figure 4 illustrates temperature isolines for stationary heat transfer problem in circular annulus with different ratios between inner and outer radii, which are in good accordance with the solution (40). Only 36 basic functions were taken with centers in knots of the square regular mesh. Numerical integration was realized by the two-dimensional method of trapezoids on the regular rectangular mesh of $32 \times 32$ nodes.

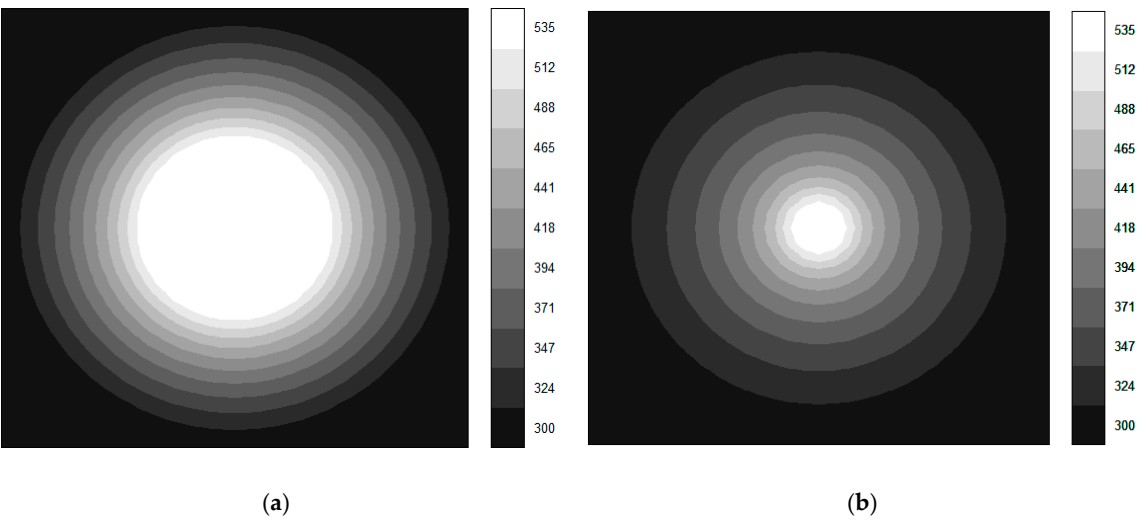

(a)                                                         (b)

**Figure 4.** Temperature isolines for different ratios: $R_i = R_o/2.6$ (**a**); $R_i = R_o/10$ (**b**).

The geometrical and thermophysical parameters were as follows:

$$R_o = 1.5 \text{ mm}; T_H = 535 \text{ K}, T_C = 300 \text{ K};$$

$$\rho = 0.54 \text{ kg/m}^3, c_p = 1000 \text{ J/(kg} \cdot K), \nu = 3.2 \cdot 10^{-5} \text{ m}^2/\text{s}, \beta = 1.5 \cdot 10^{-3} \text{ K}^{-1}.$$

Relative errors between approximate and exact solutions in **L**$_2$-norm were $\varepsilon \approx 4 \cdot 10^{-4}$ for $R_i = R_o/2.6$ and $\varepsilon \approx 7 \cdot 10^{-3}$ for $R_i = R_o/10$. Table 1 presents the relative error as a function of the number of mesh nodes. The same resuls are shown in Figure 5 in logarithmic scale.

**Table 1.** Relative errors between approximate and analytical solutions as a function of the number of mesh nodes along each direction ($R_i = R_o/10$).

| Number of Nodes | 2 | 4 | 6 | 8 | 10 | 12 |
|---|---|---|---|---|---|---|
| Relative error | 0.019 | 0.012 | $7.3 \times 10^{-3}$ | $4.3 \times 10^{-3}$ | $2.6 \times 10^{-3}$ | $1.5 \times 10^{-3}$ |

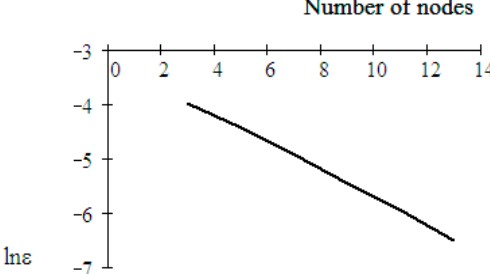

**Figure 5.** Logarithmic plot of the relative error between approximate and analytical solutions as a function of the number of mesh nodes along each direction ($R_i = R_o/10$).

Then we studied the stationary convective-diffusive heat transfer. For this set of parameters, we have $\Pr \approx 0.67$. In the case of a circular annulus, the characteristic length $L$ in the expression for the Grashof number is equal to $R_i$, i.e., $\mathrm{Gr} = \Gamma\beta(T_H - T_C)R_i^3/\nu^2$.

To increase the accuracy, instead of an expression (37) for "bonding" function $\Phi$, we can take immediately the stationary heat diffusion distribution (40):

$$\Phi(X, Y) = T_C + (T_H - T_C)\left(1 - \frac{\ln\left(R_i^{-1}\sqrt{X^2 + Y^2}\right)}{\ln(R_o R_i^{-1})}\right).$$

Figures 6 and 7 illustrate temperature and streamfunction isolines for both configurations respectively. Temperature distributions along the symmetry axis $OX$ are shown in Figure 8.

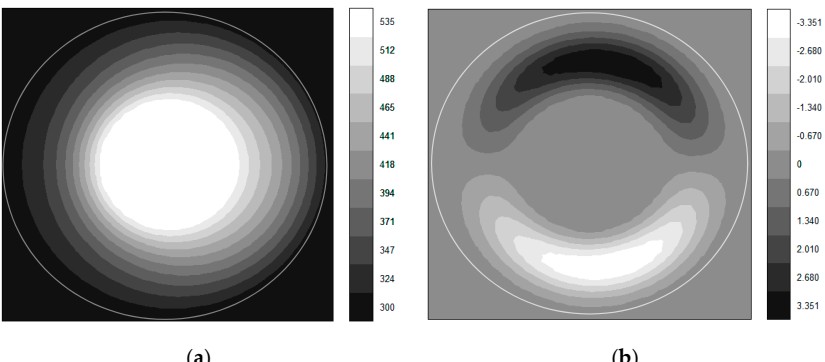

(a)  (b)

**Figure 6.** Isotherms (**a**) and streamlines (**b**) for ratio $R_i = R_o/2.6$ ($\Gamma = 3000$ g).

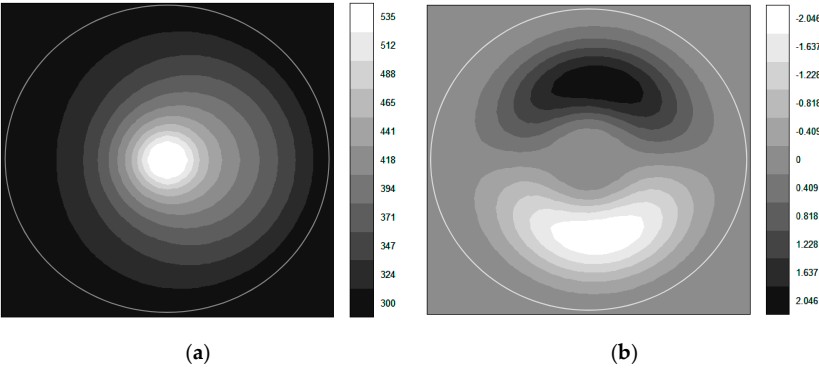

(a)  (b)

**Figure 7.** Isotherms (**a**) and streamlines (**b**) for ratio $R_i = R_o/10$ ($\Gamma = 1000$ g).

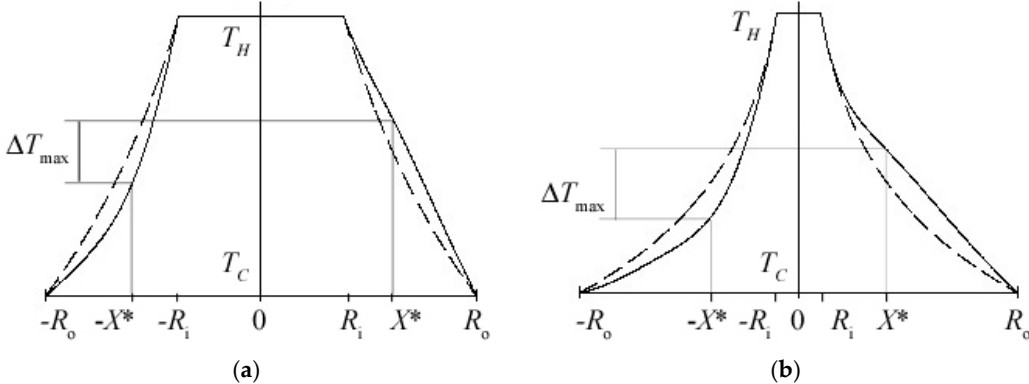

**Figure 8.** Temperature cross sections along $OX$ axis for different ratios: $R_i = R_o/2.6$ and $\Gamma = 3000$ g (solid line), $\Gamma = 0$ (dashed line) (**a**); $R_i = R_o/10$ and $\Gamma = 1000$ g (solid line), $\Gamma = 0$ (dashed line) (**b**).

Our results were compared with a well-known asymptotic solution obtained by Hodnett [24] and Mack and Bishop [25]:

$$T(r,\varphi) = T(r) + (T_H - T_C)\mathrm{Ra}f\left(\frac{r}{R_i}\right)\sin\varphi. \tag{41}$$

Here the first, "diffusive", term is defined by Equation (40) and the second, "convective", term is expressed by a rather cumbersome formula for function f with a set of coefficients presented, for example, in [24].

It observed a good correspondence between the results obtained by means of our semi-analytical approach on the base of MQ-RFM and representation (41) (within the restrictions imposed on the asymptotic solution: the inner cylinder radius $R_i$ and the temperature difference $(T_H - T_C)/T_C$ should be small enough). Our method demonstrated a very high rate of convergence. For example, in the case $R_i = R_o/2.6$, the relative error between approximate and analytical solutions became less than $1 \cdot 10^{-2}$ after only two iterations. Here, $\overline{\mathrm{Nu}}\big|_{\partial\Omega_C} \approx 242.5$, $\overline{\mathrm{Nu}}\big|_{\partial\Omega_H} \approx 238.6$, with relative errors between $1 \cdot 10^{-2}$ and $3 \cdot 10^{-2}$ respectively as compared with those evaluated with using expression (41) which gives us $\overline{\mathrm{Nu}}\big|_{\partial\Omega_C} = \overline{\mathrm{Nu}}\big|_{\partial\Omega_H} = 245.9$.

The obtained temperature profiles are also in good accordance with both experimental and numerical results obtained in the work of Garraud [36]. However, in her work, to obtain an appropriate accuracy, either a uniform finite-element grid must be used with 11,000 nodes or an adaptive non-uniform grid with 2000 nodes. At the same time, our approach provides a semi-analytical solution with the same accuracy in the form of a series of 36 terms only with respect to very simple 2D MQ-RBFs (38).

Beside temperature profiles, the temperature difference between two opposite points along the sensitivity axis $OX$ was evaluated (Figure 9) to find an optimum relative position $X* = X(\Delta T_{\max})$ of sensors of a TA with respect to the inner radius. The relative position is as follows:

$$\frac{X^* - R_i}{R_o - R_i} \approx 0.375.$$

The sensitivity S was evaluated as a function of temperature difference $\Delta T_{\max}$ from external acceleration $\Gamma = \Gamma(g)$, where g is acceleration of gravity ($g \approx 9.8$ m/s$^2$) (Figure 10). A linear part of these dependence between $\Gamma = 0$ and $\Gamma = 1000$ g corresponds to the range of "small" and "medium" accelerations and shortens as $R_i$ decreases.

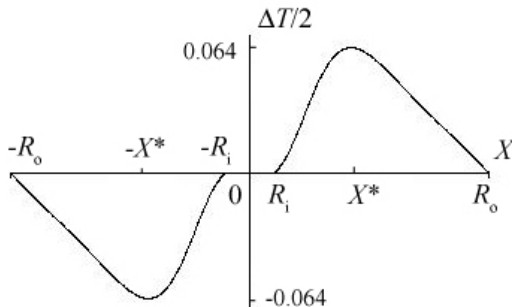

**Figure 9.** Temperature difference between two opposite points along the sensitivity axis $OX$ ($R_i = R_o/10$); optimum position corresponds to $X^*$ and $-X^*$.

It does not depend on Ra number and almost coincides with the theoretical estimation according to Hodnett's model (41) as well as to numerical and experimental data [36].

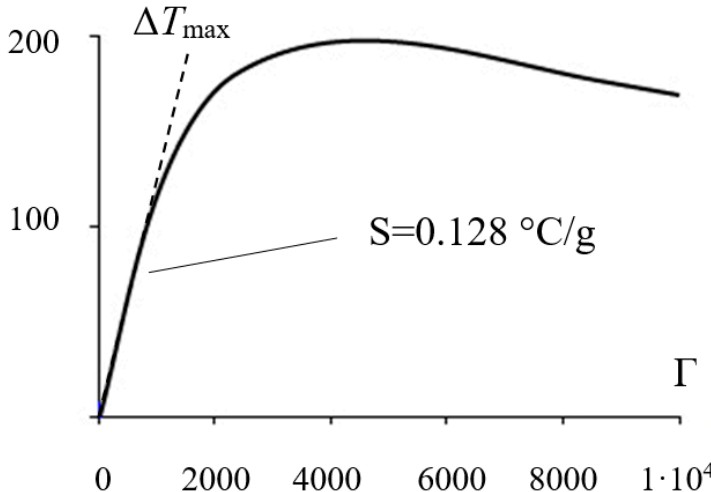

**Figure 10.** Plot of $\Delta T_{max}$ vs. external acceleration $\Gamma = \Gamma(g)$ ($R_i = R_o/10$ ); the sensitivity $S$ on the linear part between 0 and 1000 is about 0.128 °C/(m/s$^2$).

The same shape of the plots, with the steep linear slope in the range of "small" accelerations, maximum achieved at "medium" accelerations, and smooth deceleration at "high" accelerations, was observed experimentally [36]. In that work, the sensitivity S = 0.157 °C/g was measured in the interval between $\Gamma = 0$ and $\Gamma = 500$ g for the sensor with $R_o = 750$ μm, $R_i = 50$ μm. It should be noted that increase of acceleration to the values greater than some hundred g yields the turbulence phenomenon [46] which can explain partially the behavior of $S(\Gamma)$ plot. However, the simplified RFM-Galerkin scheme proposed in this article has low accuracy for such case and requires further modification to simulate turbulent flows. The results shown in Figure 10 and corresponding to values $\Gamma > 1000$ g can only be considered qualitatively.

## 7. Conclusions

The new modification of the Galerkin method for solving stationary convection-diffusion problems in arbitrarily-shaped domains was proposed. It is based on the combined use of the RFM with Boolean representation of the domain boundary and the Petrov-Galerkin iteration procedure with multiquadric RBFs. Numerical experiments showed the high accuracy and high rate of convergence of the novel approach. The semi-analytical solution was obtained in a closed form of the series with respect to MQ-RBFs and it satisfied the boundary conditions exactly. The technique was applied to the

well-studied benchmark problem of convection in the circular annulus, which is the simplest model of the thermal accelerometer. The obtained results were in good accordance with experimental data, numerical and asymptotic solutions. MQ-RFM can be applied directly for evaluation of thermal fields in more complicated domains and easily generalized to the case of other types of boundary conditions including mixed ones in different parts of the boundary. For analytical investigation of bandwidth, a transient study based on a combination of MQ-RFM and the Rothe method can be realized by analogy with [11]. As for solving 3D problems, it should be noted that, with the help of RFM it is possible to describe any 3D object geometry. However, we cannot directly transfer the technique described in the article to 3D problems due to difficulties in extending the vorticity-stream function formulation to the multidimensional case [47].

**Author Contributions:** Conceptualization, A.G.; methodology, M.B.; validation, P.C.; formal analysis, M.B.; investigation, M.B.; resources, A.G.; data curation, P.C.; writing—original draft preparation, M.B.; writing—review and editing, A.G.; visualization, M.B.; supervision, A.G.; project administration, A.G. All authors have read and agreed to the published version of the manuscript.

**Funding:** This research received no external funding.

**Conflicts of Interest:** The authors declare no conflict of interest.

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
