# Peer review of "Thermal Accelerometer Simulation by the R‑Functions Method"

_applsci, doi:10.3390/app10238373_

Round 1

Reviewer 1 Report

This paper combines the RFM and MQ method for modeling the thermal accelerometers. 

An semi-analytical solution is provided for the diffusion convection problem. Overall, this paper is well written with details background and theoretical derivation.  I have minor comments on this manuscript.

1.The labeling of some figures can be improved. For example, the Fig.4, only shows the contour of temperature with no labeling of temperature level in each figure.  In this way, the plot provide very minimum information for the readers.

2.The caption of the figure should be further improved. For example, figure 8, the parameters used for getting the solutions should be listed in the caption.

3.This works focusses on solving the 2D problems. Any comments on extending this method to 3D problems?

Author Response

The author thank the reviewer for valuable comments. The corrections are filled with yellow color in the revised version of the manuscript.

1.The labeling of some figures can be improved. For example, the Fig.4, only shows the contour of temperature with no labeling of temperature level in each figure.  In this way, the plot provide very minimum information for the readers.

ANSWER: Labeling for Fig. 4 was changed, temperature levels were added. The same was made for Fig. 6 which was split into two figures, 6 and 7. 

2.The caption of the figure should be further improved. For example, figure 8, the parameters used for getting the solutions should be listed in the caption.

ANSWER: Captions for Figs. 8 and 9 (Figs. 9 and 10 in the revised version) were improved.

3.This works focusses on solving the 2D problems. Any comments on extending this method to 3D problems?

ANSWER: Comments on extending the technique to 3D problems are given in Conclusion section together with ref. [47].

Reviewer 2 Report

The paper solves 2-D NS equation with energy equation using Vorticity-Stream function method. The method of solution sound. They used Radial base function method in solving NS V-S method. 

However, the authors did not give any justification of using their method. The results are expected. 

Author Response

The author thank the reviewer for valuable comments.

However, the authors did not give any justification of using their method. The results are expected.

ANSWER: Information about justification is given in Chapter 5 (filled with green color) together with some necessary references [41-43]:

"The detailed justification of the RFM with Galerkin technique in connection with solving the natural convection problem in enclosure regions is presented in [12, 41, 42]. In particular the natural convection in presence of local heat is investigated in [12]. This justification is based on variational principles [43] and is appropriate for the wide class of bases composed of both spectral and compactly supported functions."

41. Sidorov, M.V. Construction of structures for solving the Stokes problem. Radioelectronics and Informatics. 2002, 3, pp. 52-54. (In Russian). 

42. Sidorov, M.V. Application of R-functionsto the calculation of the Stokes flow in a square cavity at low Reynolds. Radioelectronics and Informatics. 2002, 4, p 77 - 78. (In Russian). 

43. Michlin, S.G. Variational Methods in Mathematical Physics. Pergamon Press: Oxford, 1964.

Reviewer 3 Report

The paper is interesting. However some clarifications need to be introduced and some minor corrections should be made.
The fluid velocity sholud also be specified, not just accelerations; by the fact, from the lines 303-306 the resulting Reynolds numbers would be of some hundreds and maybe more; therefore, turbulence phenomena could be triggered that would not be captured by the application of the RFM technique. Therefore the authors are invited to discuss in more detail the issue of the possible onset of turbulence. In this regard, a very important text for numerical CFD:
Chung T.J. (2006) - Computational Fluid Dynamics Cambridge University Press pp. 1012;
and, among many others, the following papers on the numerical simulation of turbulence:
Nithiarasu P., Codina R., Zienkiewicz O.C. The characteristic Based Split (CBS) scheme - a unified approach to fluid dynamics - Int. Journal for Numerical Methods in Engineering. 66, pp. 1514-1546.
Pasculli, A. (2008). CFD-FEM 2D Modelling of a local water flow. Some numerical results. Alpine and Mediterranean Quaternary. Vol. 21(B), Issue 1, 2008, pp. 215-228. ISSN: 22797327; SCOPUS: 2-s2.0-84983037047.
Lines 165 and 169: the paper is based on stationary simulation, accordingly, please change the term "time step" to "iterative step";
Line 351: Figure 9 appears to come from experimental measurements ("measured"); which unit of measurement is used for accelerations? If was selected, accordingly, it appears to be extremely high accelerations! Furthermore, why did the authors not carry out comparisons between the results of the experimental measurements (if they were such) and the numerical results?
Line 362: only non-stationary phenomenology was discussed in the paper, therefore the term "non-stationary" must be deleted.
line 104, typo: please change "desigm" to "design". 

Author Response

The authors thank the reviewer for valuable comments. The corresponding corrections are filled with blue color in the revised version of the manuscript.

The fluid velocity should also be specified, not just accelerations; by the fact, from the lines 303- 306 the resulting Reynolds numbers would be of some hundreds and maybe more; therefore, turbulence phenomena could be triggered that would not be captured by the application of the RFM technique. Therefore the authors are invited to discuss in more detail the issue of the possible onset of turbulence.

ANSWER: This question is very interesting and is considered from the perspective point of view mainly. It should be noted that increase of acceleration to the values greater than some hundred g yields the turbulence phenomenon which can explain partially the behavior of S(Γ) plot. However, the simplified RFM-Galerkin scheme proposed in this article has low accuracy for such case and requires further modification to simulate turbulent flows. The results shown in Fig. 10 (Fig. 9 in original version) and corresponding to values Γ>1000g can only be considered qualitatively.

Lines 165 and 169: the paper is based on stationary simulation, accordingly, please change the term "time step" to "iterative step".

ANSWER: The authors agree, the term “time step” was incorrect; the corrections were made (Lines 167 and 171 in the revised version).

Line 351: Figure 9 appears to come from experimental measurements ("measured"); which unit of measurement is used for accelerations? If was selected, accordingly, it appears to be extremely high accelerations! Furthermore, why did the authors not carry out comparisons between the results of the experimental measurements (if they were such) and the numerical results?

ANSWER: Incorrect “measured” was changed for “evaluated”. The unit of measurement used for acceleration is g, m/s2. Comparison with close results of A. Garraud [36] is added.

Line 362: only non-stationary phenomenology was discussed in the paper, therefore the term "non-stationary" must be deleted.

ANSWER: The authors agree, the term “non-stationary” was deleted (Line 383 in the reviewed version).

line 104, typo: please change "desigm" to "design".

ANSWER: The error was corrected (Line 105 in the reviewed version).